# Copper-catalysed asymmetric reductive cross-coupling of prochiral alkenes

Wan Seok Yoon[1,4], Won Jun Jang[1,3,4], Woojin Yoon[2], Hoseop Yun [2✉] & Jaesook Yun [1✉]

Asymmetric construction of $C(sp^3)$–$C(sp^3)$ bond with good stereocontrol of the two connecting carbon centres retaining all carbon or hydrogen substituents is a challenging target in transition metal catalysis. Transition metal-catalysed reductive coupling of unsaturated π-substrates is considered as a potent tool to expediently develop the molecular complexity with high atom efficiency. However, such an asymmetric and intermolecular process has yet to be developed fully. Herein, we report an efficient strategy to reductively couple two prochiral conjugate alkenes using a copper-catalysed tandem protocol in the presence of diboron. Notably, this transformation incorporates a wide range of terminal and internal enynes as coupling partners and facilitates highly diastereo- and enantioselective synthesis of organoboron derivatives with multiple adjacent stereocentres in a single operation.

[1] Department of Chemistry and Institute of Basic Science, Sungkyunkwan University, Suwon 16419, Korea. [2] Department of Energy Systems Research and Department of Chemistry, Ajou University, Suwon 16499, Korea. [3] Present address: Department of Chemistry, Dong-A University, Busan 49315, Korea. [4] These authors contributed equally: Wan Seok Yoon, Won Jun Jang. ✉email: hsyun@ajou.ac.kr; jaesook@skku.edu

Alkenes are easily accessible feedstocks in synthetic organic chemistry, serving as versatile precursors in a number of transition metal-catalysed processes as well as classical organic reactions. While traditional metal-catalysed cross-coupling of pre-functionalised coupling partners using Pd and Ni-catalysts has been widely recognised for the formation of C–C bonds[1,2], transition-metal catalysed reductive coupling of unsaturated π-substrates[3,4], is a possible and potent means to develop complex molecules with high atom efficiency.

Asymmetric construction of C–C(sp$^3$) bonds is of importance and represents an intriguing research target in transition-metal catalysis[5,6]. While various synthetic methodologies have been developed, asymmetric formation of C(sp$^3$)–C(sp$^3$) bond with concomitant stereocontrol of the two connecting carbon centres carrying all carbon or hydrogen substituents is still a formidable challenge (Fig. 1a)[7,8]. Such a challenging task was recently achieved by the Fu group reporting on Ni-catalysed enantio-convergent radical coupling with racemic secondary silylated propargyl bromides and alkyl zinc nucleophiles (secondary alkyl-alkyl coupling) (Fig. 1b)[9]. Intermolecular reductive coupling of

two prochiral alkenes represents an alternative approach to address stereochemical challenges involving the generation of two stereocentres of the connecting C(sp$^3$)–C(sp$^3$) bond. However, such studies of transition-metal catalysed intermolecular reductive coupling of two alkenes are scarce, since most successful methodologies reported so far utilised alkene-derived nucleophiles with C=O and C=N acceptors[10,11]. In 2014, Baran and coworkers pioneered the iron-catalysed olefin cross-couplings in which olefin-derived radical intermediates through a HAT/radical recombination were captured by Michael acceptors, resulting in reductive coupling of two different alkenes (Fig. 1b)[12,13]. However, asymmetric versions of these protocols have yet to be established. Hence, the development of asymmetric catalytic methods that employ two prochiral alkenes to form C(sp$^3$)–C(sp$^3$) bonds with concomitant stereocontrol is a worthwhile research target.

Recently, our laboratory and others reported copper-catalysed conjugative addition of alkylcopper[14,15] or allylcopper[16,17] nucleophiles to Michael acceptors using corresponding organoboron substrates or dienes, without using strongly basic and

**a** Asymmetric induction at both carbons of the new C(sp$^3$)–C(sp$^3$) bond

**b** Prior works (radical involved)

preformed reagents

Fu group (2020), *Science*

racemic examples only

Baran group (2014), *Nature*

**c** Copper-catalysed conjuagate addition of chiral alkyl nucleophiles with α-heterofunctionality

catalytic

**d** Copper-catalysed reductive coupling of two conjugate alkenes (this work)

chiral copper catalyst

L*Cu-B

diastereo- and enantioselective

**Fig. 1 Overview of C(sp$^3$)–C(sp$^3$) bond formation. a** Asymmetric induction of the new C(sp$^3$)–C(sp$^3$) bond at both carbons. **b** Prior works (radical involved). **c** Copper-catalysed conjugate addition of chiral alkyl nucleophiles with α-heterofunctionality. **d** Copper-catalysed reductive coupling of two conjugate alkenes (this work).

preformed nucleophiles such as Grignard and organolithium reagents. However, all those methods formed an alkyl-alkyl bond with a single stereocentre and especially, the addition of allyl copper species derived from dienes resulted in a mixture of (E) and (Z)-isomers[16]. Moreover, further attempts to increase the steric bulk of the alkylcopper-nucleophile derivatives from primary to secondary were not successful. While conjugate addition of a few configurationally stable chiral alkyl metallic reagents with α-heterofunctionality as the nucleophiles[18,19] and 1,6-conjugate addition of allene-derived nucleophiles[20] represented successful conjugate reactions, we recently reported that catalytic alkenylboron-derived copper nucleophiles could be utilized to accomplish diastereo- and enantioselective conjugate addition of α-borylalkyl copper species (Fig. 1c)[21]. Inspired by this work, we envisioned a non-radical strategy for asymmetric assembly of two prochiral alkenes with stereocentre formation between the two connecting carbons bearing no heteroatom substituents (Fig. 1d). The challenges include chemoselectivity between two prochiral alkenes toward an activated copper catalyst and reactivity of an intermediate copper nucleophile with the second alkene.

Here, we show the realization of coupling processes using enynes and Michael acceptors in the presence of a copper catalyst. The protocol accommodates both terminal and internal enynes and allows reductive coupling of two alkene moieties in the presence of diboron in a multicomponent and tandem fashion, resulting in molecular complexity with formation of two to three stereogenic centres in a single operation.

## Results and discussion

**Reaction optimisation for coupling terminal enynes.** Conjugate olefins such as enynes[22–24] and alkylidene diesters are both reactive electrophiles towards nucleophilic copper-catalysts. In order to address the chemoselectivity issue of two reacting olefins, we initiated our optimisation studies by investigating the reaction between terminal enyne **1a** and benzylidene malonate **2a** with B₂pin₂ as a stoichiometric reductant in the presence of copper catalyst under various conditions (Fig. 2). Surprisingly, the copper catalyst combined with a bisphosphine ligand **L1** reductively coupled two conjugate alkenes to yield racemic **3a** in high yield with high diastereoselectivity (>98:2). Encouraged by the result, various chiral electron-donating ligands and copper precursors were further screened to optimise reaction conditions for the borylative coupling for **3a** with high yield and enantioselectivity. Among the ligands screened, the Josiphos ligand (**L6**) was found suitable and yielded the desired product in with high diastereo- and enantioselectivity. Use of other copper precursors was less efficient, and NaOt-Bu instead of LiOt-Bu yielded no product presumably due to reduced Lewis acidity of Na⁺ [25,26]. Enantioselectivity was further improved to 93% ee by decreasing the reaction temperature to 0 °C, which was selected as the optimal condition.

**Substrate scope.** Next, the scope of the olefin cross-coupling was investigated (Fig. 3). The chiral copper catalyst combined with ligand **L6** efficiently coupled enyne **1a** with various β-substituted alkylidene malonates **2** in the presence of B₂pin₂. This three-component coupling reaction displayed a broad scope with respect to alkylidene malonate **2** carrying aromatic, heteroaromatic and alkyl groups, resulting in the formation of the coupled products **3a**–**3o** with good diastereoselectivity, enantioselectivity and yield. While the yield and enantioselectivity was not highly sensitive to the β-substituent of the malonate **2**, a slight decrease in yield was observed for sterically hindered o-tolyl and 2° alkyl substituents. Notably, the 1,4-addition product (**3o**) is preferred over 1,6-addition product when a α,β,γ,δ-dienoate was used. We then analysed the substrate scope of the terminal enyne **1** with **2a**. Electron-rich aromatic substituted enynes were successfully transformed to the corresponding coupled product with high enantioselectivity. However, enynes with a highly electron-withdrawing or ortho-substituted aromatic groups, such as −CF₃, −CO₂Me and o-tolyl, afforded product with low-to-moderate ee values under condition A. However, those substrates bearing an electron-deficient aromatic group showed

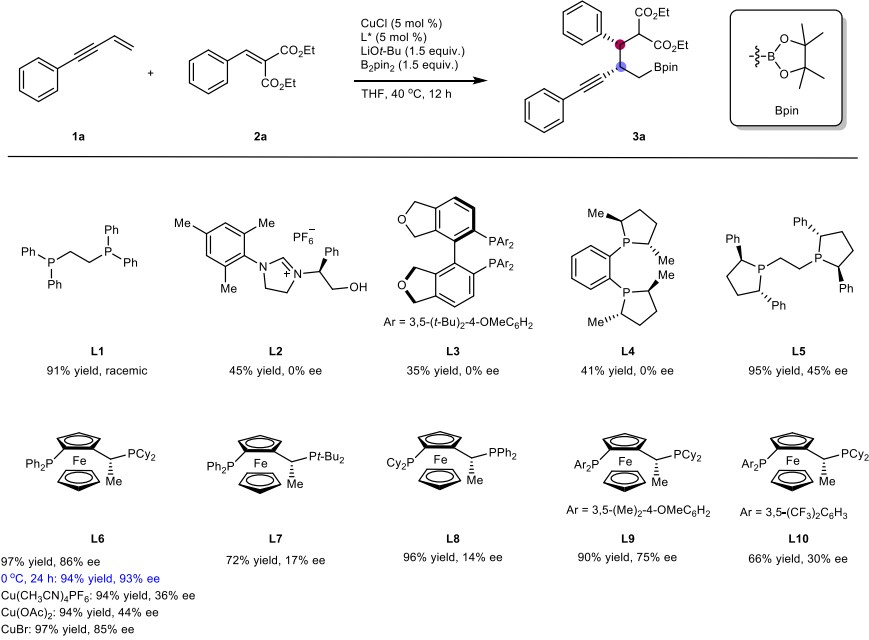

**Fig. 2 Optimisation of coupling reaction conditions.** All reactions were carried out in a 0.5 mmol scale using **1a** (1.5 equiv.) and **2a** (1.0 equiv.) under a N₂ atmosphere at 40 °C, unless otherwise noted. Yields of **3a** are isolated and diastereomeric ratio of product **3a** was determined via ¹H NMR analysis of the unpurified mixtures and was high (>98:2) in all cases. B₂pin₂ = Bis(pinacolato)diboron.

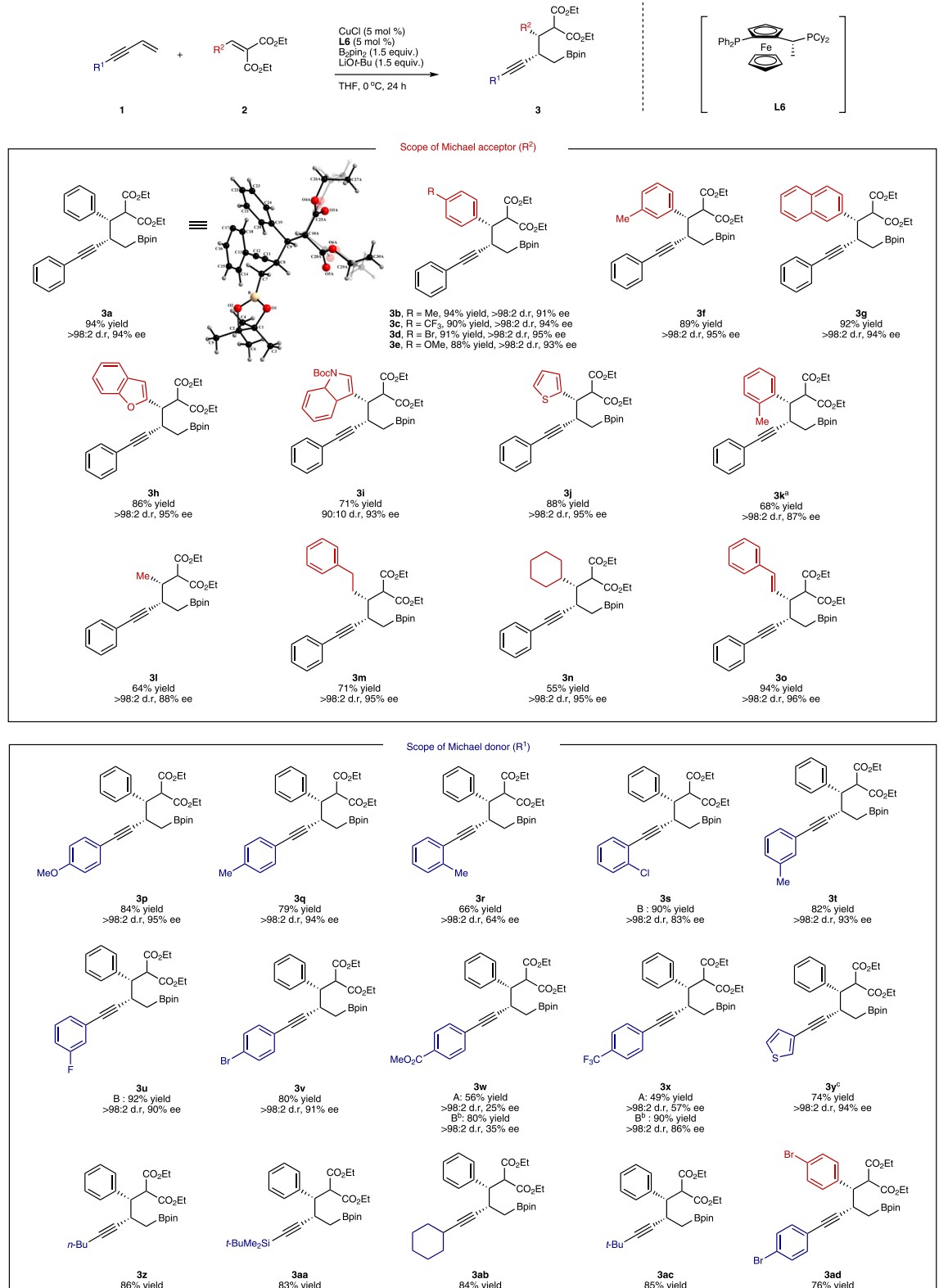

**Fig. 3 Substrate scope of terminal enynes and alkylidene malonates.** Reactions were generally conducted in a 0.5 mmol scale using **1** (1.5 equiv.) and **2** (1.0 equiv.) at 0 °C (condition A) unless otherwise noted. Under condition B, **1** (1.0 equiv.) and **2** (2.0 equiv.) were used at 0 °C. a The reaction was carried out at 40 °C. b The reaction was carried out in methyl tert-butyl ether (MTBE) instead of THF. c 1.5 equiv. of **2** was used.

**Fig. 4 Substrate scope of internal enynes.** Reactions were generally conducted in a 0.5 mmol scale using **4** (1.2 equiv.) and **2** (1.0 equiv.) with the **L5**-copper catalyst at 40 °C unless otherwise noted. a %ee was determined with **6 f**. b MTBE was used as solvent.

increased yield and enantioselectivity when using two-fold quantities of Michael acceptors **2** relative to **1** (condition B). Overall, the variation of substituent (R$^1$) of the terminal enyne **1** affected the reaction substantially more than the variation of substituent (R$^2$) of **2** in these olefin cross-coupling reactions.

**Reductive coupling of internal enynes**. The chiral Ph-BPE ligand (**L5**) was optimal for controlling three stereogenic centres with **2a** instead of **L6** when more challenging internal enynes were used. (Z)-Enyne (**4a**) was more suitable than (E)-enyne for the formation of addition product **5a** with good enantio- and diastereoselectivity; a single diastereomeric product was formed with high enantioselectivity exceeding 99% ee. The chiral copper-**L5** catalyst at 40 °C coupled methyl and primary alkyl substituted internal enynes with benzylidene malonates (**2**) to afford product **5** with concomitant formation of three stereogenic centres with high stereocontrol in a single operation (Fig. 4). However, ethylidene malonate was less efficient in the coupling with internal enynes to yield **5n** possibly due to competing deconjugation of the malonate[26] and a secondary alkyl-substituted enyne failed to afford the desired product (**5o**).

**Additional data and applications**. To gain some insight into the reaction, Hammett studies were carried out (Fig. 5a). A Hammett plot with terminal enynes (**1**) carrying a different aromatic substituent under the condition A in Fig. 3 showed a linear correlation with a negative slope ($\rho = -0.51$)[27]. The results suggested that copper intermediates formed from electron-rich enynes react more efficiently with benzylidene malonate (**2**) than those from electron-deficient enynes (**1v**), minimising racemisation at the propargylic-copper stereogenic centre up to halogen-substituted substrates. Further, the increased yield of **3w** and **3x** containing strongly electron-withdrawing substrates with excess Michael acceptors under condition B in Fig. 3 indicated the nucleophilic character of an intermediate organocopper species. We also carried out a coupling reaction in the presence of TEMPO (2,2,6,6-tetra-methyl-1-piperidinyloxy) with no significant decline in yield or ee, and no TEMPO adducts, indicating the absence of significant role of organic radicals in this process (see the Supplementary Information for details)[28]. Our proposed catalytic cycle is illustrated in Fig. 5b, which comprises chemoselective and enantioselective syn-addition of a ligand-coordinated Cu–Bpin catalyst to enyne[29] and subsequent conjugate

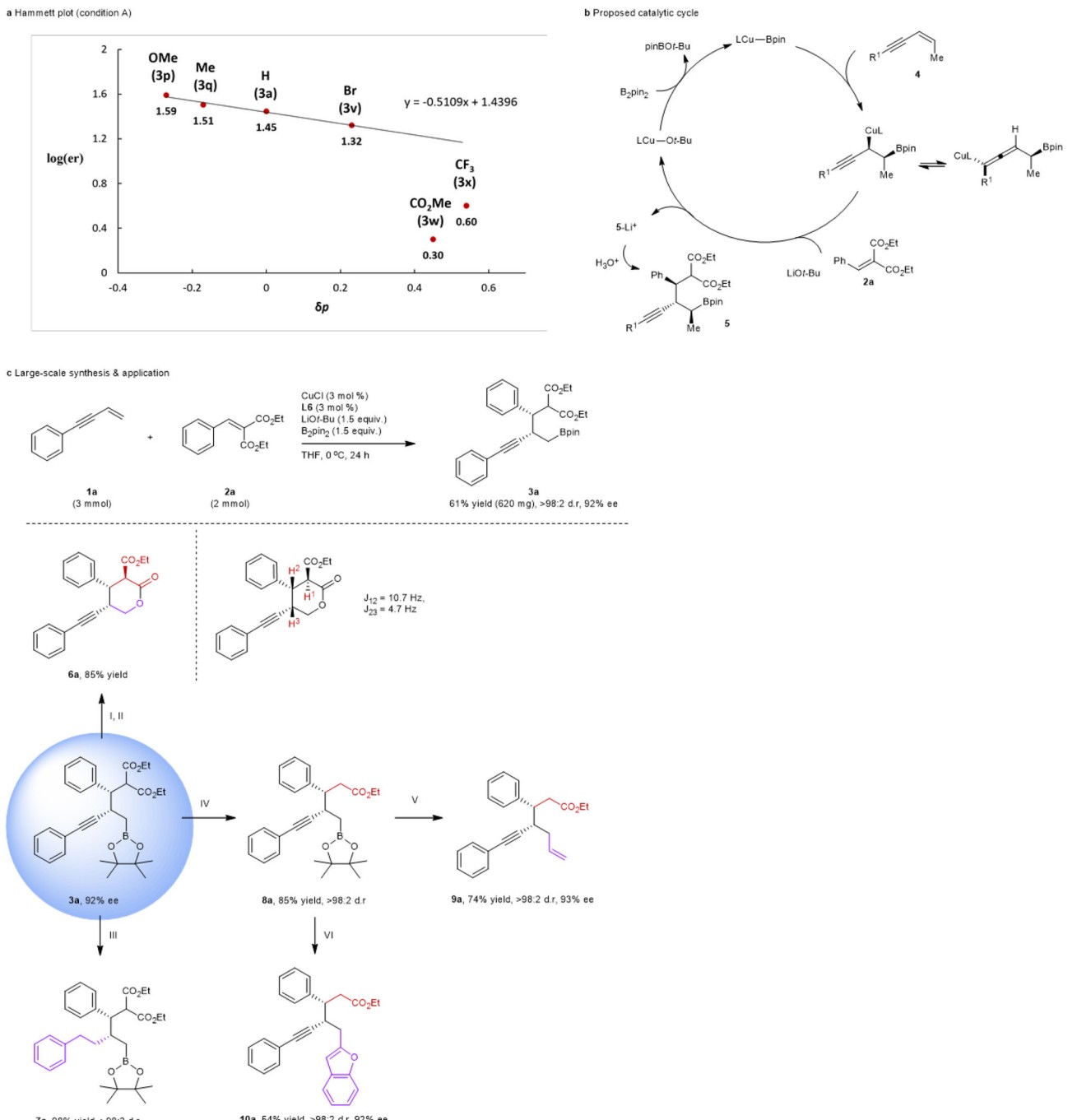

**Fig. 5 Mechanistic data and applications. a** Hammett plot (condition A). **b** Proposed catalytic cycle. **c** Large-scale synthesis & application. Reaction conditions: I. NaBO₃, H₂O/THF, rt. II. *p*-TSOH, benzene, rt. III. Pd/C, H₂, THF, rt. IV. NaCl, DMSO/H₂O, 160 °C. V. Vinylmagnesium bromide, I₂, THF, MeOH −78 °C. VI. Benzofuran, *n*-BuLi, NBS, THF, −78 °C.

addition of a generated copper intermediate with benzylidene malonate in the presence of a lithium alkoxide base. The copper alkoxide generated undergoes σ-bond metathesis with diboron to regenerate the Cu–Bpin catalyst, completing the catalytic cycle. The lithium enolate of the malonate adduct yields the desired product during work-up. A gram scale synthesis of **3a** conducted with 3 mol % catalyst and ligand provided the desired product, but with slightly decreased yield and ee (Fig. 5c). Lastly, we briefly examined organic transformations of the resulting coupled products obtained from these olefin cross-coupling reactions to demonstrate the potential

synthetic power of the current methodology (Fig. 5c). First, the coupled product **3a** was transformed into a useful enantio-enriched lactone derivative[30]. Oxidation of the Bpin group and cyclisation yielded highly substituted, chiral δ-lactone **6a** as a single stereoisomer. Notably, the lactonisation created another stereogenic centre and ¹H coupling constants (³J) matched with the relative configuration reported in the literature (Fig. 5c)[31]. Next, hydrogenation of the triple bond of **3a** produced a reduced alkylboronate ester **7a** and decarboxylation resulted in the formation of **8a**. Further conversion of the C–B bond of **8a** to C–C bonds by vinylation and benzofuranylation

yielded desired products **9a** and **10a** without a change in enantiomeric excess.

In summary, our strategy effectively generates two stereocentres of the connecting C(sp$^3$)–C(sp$^3$) bond via intermolecular reductive coupling of two prochiral conjugate alkenes. This process accommodates a wide range of terminal and internal enynes as coupling partners and enables highly stereoselective synthesis of organoboron derivatives and lactones bearing adjacent stereocentres. The scope of these reductive olefin cross-coupling reactions including less activated alkenes as well as conjugated acceptors based on mechanistic studies will be expanded and reported in due course.

## Methods

**General procedure for the reductive coupling of 1 and 2**. A mixture of CuCl (5 mol %, 0.025 mmol), **L6** (5 mol %, 0.025 mmol), LiO$t$-Bu (1.5 equiv, 0.75 mmol), and B$_2$pin$_2$ (1.5 equiv., 0.75 mmol) in THF (0.7 mL) was stirred for 15 min in a Schlenk tube under an atmosphere of nitrogen. Substrate **1** (1.5 equiv., 0.75 mmol) and **2** (1 equiv., 0.5 mmol) dissolved in THF (0.3 mL) were added to the reaction mixture at 0 °C. The reaction mixture was stirred at 0 °C and monitored by TLC. Upon complete consumption of **2**, the reaction mixture was diluted with water (3 mL) and extracted with dichloromethane (5 mL x 3). The combined organic layers were washed with brine, dried over MgSO$_4$, and concentrated in vacuo. The residue was purified by column on silica gel using ethyl acetate/hexanes as eluent.

## Data availability

Materials and methods, experimental procedures, optimisation studies, $^1$H NMR spectra and $^{13}$C NMR spectra data are available in the Supplementary Information. Additional data are available from the corresponding author upon request. The X-ray crystallographic data of compound **3a** and **5m** have been deposited at the Cambridge Crystallographic Data Centre (CCDC) with the accession code CCDC 2121298 (**3a**) and CCDC 2121073 (**5m**) (https://www.ccdc.cam.ac.uk/structures/).

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

## Acknowledgements

This work was supported by Samsung Science and Technology Foundation under Project Number SSTF-BA2002-08 (to J.Y.) X-ray structural analysis was supported by Basic Science Research Program through the National Research Foundation of Korea (2019R1I1A2A01058066, to H.Y.).

## Author contributions

W.S.Y. and W.J.J. optimised reaction conditions and expanded substrate scope. W.S.Y. prepared crystals for X-ray analysis. W.Y. carried out X-ray analysis under the guidance of H.Y. J.Y. led the project. W.S.Y. and J.Y. wrote the manuscript with contributions from all authors.

## Competing interests

The authors declare no competing interests.
