## [Peer Review File · Nature Communications]

REVIEWER COMMENTS

Reviewer #1 (Remarks to the Author):

The authors described a copper-catalysed asymmetric reductive addition of 1,3-enynes to α,β -unsaturated carbonyl compounds, which resulted in the formation of consecutive chiral tertiary carbon centers. It is impressive that the substrate scope was broad and both the diastereoselectivity and the enantioselectivity were excellent. However, the challenges described in the manuscript have been well addressed previously in ref 17. Therefore, the novelty of the present manuscript does not meet the high requirement of Nature Communications. Thus the submission to a specialized journal is highly recommended. Moreover, some issues should be addressed before its next submission.

1. 1,3-Enynes rather than alkenes should be well documented. Especially, the examples on the reductive or borylative cross-coupling between 1,3-enynes and aldehydes/ketones could be cited.
2. The authors discussed the challenge in the asymmetric formation of C(SP³)-C(SP³) bond with concomitant stereocontrol of the two connecting carbon centers. Actually, such consecutive chiral tertiary carbon centers were well produced by the Michael addition of α -substituted enolates (and analogs) to α,β -unsaturated carbonyl compounds. Thus the Figure 1a should be redrawn.
3. It is well known that such copper-catalysed borylative cross-couplings do not proceed in a radical-based mechanism. Thus it looks like that there is no need to mention these radical examples in Figure 1b.
4. Since there are many catalytic asymmetric examples on the copper (or other metals)-catalysed conjugate addition of chiral alkyl nucleophiles with α -heterofunctionality (such as catalytic asymmetric Michael addition with α -imino-esters), the example with stoichiometric chiral reagent shown in figure 1c could be replaced.
5. The transformations did not fully make use of the C-BPin group. Both C-C bond formation and C-X bond formation (other than oxidation) with C-BPin group would strengthen the present manuscript significantly.

Reviewer #2 (Remarks to the Author):

Yun and co-worker describe a Cu-catalyzed reductive protocol for three-component coupling of enynes, alkylidene malonates, and B₂(pin)₂. The use of a common chiral copper catalyst has enabled modular access to valuable organoboron derivatives with good stereocontrol of two or three consecutive, sp³-hybridized carbon stereogenic centers. A wide variety of structurally diverse enynes and alkylidene malonates can be readily incorporated into this mild protocol to generate the products in good yields and with excellent diastereo- and enantioselectivities. Although the previous work (ref. 17) by Hoveyda might somewhat compromise the conceptual novelty, this study takes a few steps forward to rapidly access notoriously difficult enantioenriched sp³ linkages from simple starting materials with improved chiral complexity and structural modularity. This work represents an important advance in the field of asymmetric catalysis and provides a practical complementary method to traditional transition metal-catalyzed cross-coupling reactions to address the long-standing challenging goal of enantioselective C(sp³)-C(sp³) bond constructions. As such, I strongly support the publication of this work in Nature Communications.

Some minor points to be considered as below:

1. Could the authors please comment to the substantial difference between tBuOLi and tBuONa for the reaction outcome? In Hoveyda's work (ref. 17), tBuONa was shown to be a competent base for a relevant transformation.

2. It would be interesting to know whether Hoveyda's reaction conditions could be employed the outlined transformations in this work and whether enynes could be replaced with 1,3-dienes in this study to access related asymmetric transformations.
3. If a common hydrosilane was used as the stoichiometric reductant instead of B₂(pin)₂, could the reductive alkyl-alkyl coupling products be achieved?
4. The demonstration of the scalability of this method may further improve the utility.
5. A typo: The term of "two-fold excess" should be changed to "two-fold quantities".
6. A mechanistically relevant paper is suggested to be cited (Angew. Chem. Int. Ed. 2011, 50, 2778–2782).
7. The obvious impurities are found in the NMR spectra of compound 3l.
8. Errors of page/year numbers are found in ref. 10 and ref.11.

Reviewer #3 (Remarks to the Author):

Yoon and co-workers reported a method for asymmetric reductive coupling using enynes and Michael acceptors to generate products with valuable multiple stereocenters in high selectivities. The method presented a broad scope, and group tolerance. After addressing all these revisions, I recommended publish in Nature Communications.

1. It would make the figures look better if the conditions and the arrows align in the middle, as well as keep the vertical space equal.
2. In the sentence, "...concomitant stereocontrol is worthwhile research target", should it be "...concomitant stereocontrol is a worthwhile research target"?
2. In the sentence, " Use of other copper precursors was less efficient...", should it be "...were less efficient..."?
3. In Fig. 4a, the Hammett plot has already included the information that listed in the table, which seems repetitive in the paper, and moving the table to Supporting Information is highly recommended.
4. In Fig. 4b, the author has shown the possibility of generating the Cu-allene intermediate, was the intermediate able to be monitored during the reaction?
5. It would be more helpful to correlate by Fig. 4d in the paragraph that describes the application.
6. The authors did two similar applications in Fig. 4d, are there any other possible product derivatizations that can be done with this type of molecules? For example, cyclization from the alkyne functional group of compound 6a to form a tricyclic molecule?
7. In the conclusion and outlook part, the authors should give some viewpoints where the field will grow and develop in the near future.
8. The format of the reference should be consistent.
 - a. missing a period in Ref.3
 - b. some references using a range of the paper, some using the starting page.

REVIEWER COMMENTS

Reviewer #1 (Remarks to the Author):

The authors described a copper-catalysed asymmetric reductive addition of 1,3-enynes to α,β -unsaturated carbonyl compounds, which resulted in the formation of consecutive chiral tertiary carbon centers. It is impressive that the substrate scope was broad and both the diastereoselectivity and the enantioselectivity were excellent. However, the challenges described in the manuscript have been well addressed previously in ref 17. Therefore, the novelty of the present manuscript does not meet the high requirement of Nature Communications. Thus the submission to a specialized journal is highly recommended. Moreover, some issues should be addressed before its next submission. → replied to the Editor's request.

1. 1,3-Enynes rather than alkenes should be well documented. Especially, the examples on the reductive or borylative cross-coupling between 1,3-enynes and aldehydes/ketones could be cited. → Ref 11 in the original version contained such examples. In this revision, we changed the position and added another reference. (ref 22 and 23)

2. The authors discussed the challenge in the asymmetric formation of C(SP³)-C(SP³) bond with concomitant stereocontrol of the two connecting carbon centers. Actually, such consecutive chiral tertiary carbon centers were well produced by the Michael addition of α -substituted enolates (and analogs) to α,β -unsaturated carbonyl compounds. Thus the Figure 1a should be redrawn. → Enolate chemistry that includes deprotonation of an acidic \$\alpha\$ proton of carbonyls is not relevant to our current topic. Generation of chiral alkyl metallic species(nucleophiles) from non-participating or less acidic substrates is challenging and is the goal of our investigation. Traditionally, chiral alkyl lithium reagents with coordinating \$\alpha\$ -heterofunctionality via deprotonation of benzylic proton are popular in conjugate addition. Anyhow, we removed the-lithium related scheme to focus on \$\pi\$ -substrates. The example of \$\alpha\$ -imino-esters given by the reviewer (comment 4) also used the acidity of the \$\alpha\$ proton of the ester, which happened to have \$\alpha\$ -heterofunctionality.

3. It is well known that such copper-catalysed borylative cross-couplings do not proceed in a radical-based mechanism. Thus it looks like that there is no need to mention these radical examples in Figure 1b. → this work reports the first conjugate addition of an organocopper intermediate derived from enynes and we thought it was necessary to check whether the reaction proceeds in a radical pathway or not. Some borylative substitutions take place on a radical mechanism. We left the comment of the work in the text, but detailed reaction schemes were moved to the Supporting Information (IV).

4. Since there are many catalytic asymmetric examples on the copper (or other metals)-catalysed conjugate addition of chiral alkyl nucleophiles with α -heterofunctionality (such as catalytic asymmetric Michael addition with α -imino-esters), the example with stoichiometric chiral reagent shown in figure 1c could be replaced. → As mentioned in our response to the comment 2, the example of \$\alpha\$ -imino-esters is silver-enolate chemistry with an imino N-substituent. We added this reference as ref 19 as an example of chiral alkyl metal with \$\alpha\$ -heterofunctionality.

5. The transformations did not fully make use of the C-BPin group. Both C-C bond formation and C-X bond formation (other than oxidation) with C-BPin group would strengthen the present manuscript significantly. → other applications in Fig 4c.

Reviewer #2 (Remarks to the Author):

Yun and co-worker describe a Cu-catalyzed reductive protocol for three-component coupling of enynes, alkylidene malonates, and B₂(pin)₂. The use of a common chiral copper catalyst has enabled modular access to valuable organoboron derivatives with good stereocontrol of two or three consecutive, sp³-hybridized carbon stereogenic centers. A wide variety of structurally diverse enynes and alkylidene malonates can be readily incorporated into this mild protocol to generate the products in good yields and with excellent diastereo- and enantioselectivities. Although the previous work (ref. 17) by Hoveyda might somewhat compromise the conceptual novelty, this study takes a few steps forward to rapidly access notoriously difficult enantioenriched sp³ linkages from simple starting materials with improved chiral complexity and structural modularity. This work represents an important advance in the field of asymmetric catalysis and provides a practical complementary method to traditional transition metal-catalyzed cross-coupling reactions to address the long-standing challenging goal of enantioselective C(sp³)-C(sp³) bond constructions. As such, I strongly support the publication of this work in Nature Communications.

Some minor points to be considered as below:

1. Could the authors please comment to the substantial difference between tBuOLi and tBuONa for the reaction outcome? In Hoveyda's work (ref. 17), tBuONa was shown to be a competent base for a relevant transformation. → we added our comment to page 3 and added red 25 and 26.
2. It would be interesting to know whether Hoveyda's reaction conditions could be employed the outlined transformations in this work and whether enynes could be replaced with 1,3-dienes in this study to access related asymmetric transformations. → Our original manuscript contained those information. L₂ ligand and NaOt-Bu were the reagents used in Hoveyda's conditions. However, both reagents were ineffective for our catalysis in view of enantioselectivity and reactivity (no reaction)..
3. If a common hydrosilane was used as the stoichiometric reductant instead of B₂(pin)₂, could the reductive alkyl-alkyl coupling products be achieved? → Chemoselectivity problem occurred and reductive alkyl-alkyl coupling did not took place with our substrates combination. With a silane (TMDSO), we observed no coupled product formation, but observed reduced product of benzylidene malonate 2a as major.
4. The demonstration of the scalability of this method may further improve the utility. → Fig 4c
5. A typo: The term of "two-fold excess" should be changed to "two-fold quantities". → "two-fold quantities", not qualities
6. A mechanistically relevant paper is suggested to be cited (Angew. Chem. Int. Ed. 2011, 50, 2778–2782). → added as ref. 29.
7. The obvious impurities are found in the NMR spectra of compound 3l. → replaced by a clean NMR of 3l in the SI
8. Errors of page/year numbers are found in ref. 10 and ref.11. → all corrected.

Reviewer #3 (Remarks to the Author):

Yoon and co-workers reported a method for asymmetric reductive coupling using enynes and Michael acceptors to generate products with valuable multiple stereocenters in high selectivities. The method presented a broad scope, and group tolerance. After addressing all these revisions, I recommended publish in Nature Communications.

1. It would make the figures look better if the conditions and the arrows align in the middle, as well as keep the vertical space equal. → done
2. In the sentence, "...concomitant stereocontrol is worthwhile research target", should it be "...concomitant stereocontrol is a worthwhile research target"? → corrected
2. In the sentence, " Use of other copper precursors was less efficient...", should it be "...were less efficient..."? → the subject of this sentence is 'use'. I guess 'was' is right.
3. In Fig. 4a, the Hammett plot has already included the information that listed in the table, which seems repetitive in the paper, and moving the table to Supporting Information is highly recommended. → The Hammett plot table was transferred to SI
4. In Fig. 4b, the author has shown the possibility of generating the Cu-allene intermediate, was the intermediate able to be monitored during the reaction? → We tried to observe the Cu-allene intermediate by ¹H NMR, but couldn't verify it. However, depending on electrophiles, there are some allenyl product formations reported in the literature, which suggests its intermediacy.
5. It would be more helpful to correlate by Fig. 4d in the paragraph that describes the application. → done in page 8.
6. The authors did two similar applications in Fig. 4d, are there any other possible product derivatizations that can be done with this type of molecules? For example, cyclization from the alkyne functional group of compound 6a to form a tricyclic molecule? → application in Fig 4c.
7. In the conclusion and outlook part, the authors should give some viewpoints where the field will grow and develop in the near future. → last sentence in the summary.
 - a. missing a period in Ref.3 → corrected
 - b. some references using a range of the paper, some using the starting page. → unified pages into the starting page.

REVIEWERS' COMMENTS

Reviewer #2 (Remarks to the Author):

The authors have addressed all my previous comments in a satisfactory fashion. This manuscript is ready to go.

Reviewer #3 (Remarks to the Author):

Yun and co-workers have replied to my original comments. I support the publication of this revised version in Nature Communications.